# Evaluation of the Targeting and Therapeutic Efficiency of Anti-EGFR Functionalised Nanoparticles in Head and Neck Cancer Cells for Use in NIR-II Optical Window

**DOI:** 10.3390/pharmaceutics13101651

**Published:** 2021-10-09

**Authors:** Teklu Egnuni, Nicola Ingram, Ibrahim Mirza, P. Louise Coletta, James R. McLaughlan

**Affiliations:** 1Leeds Institute of Medical Research, University of Leeds, St. James’s University Hospital, Leeds LS9 7TF, UK; T.Egnuni@leeds.ac.uk (T.E.); N.Ingram@leeds.ac.uk (N.I.); P.L.Coletta@leeds.ac.uk (P.L.C.); 2School of Electronic and Electrical Engineering, University of Leeds, Leeds LS2 9JT, UK; elinm@leeds.ac.uk

**Keywords:** gold nanorods, targeting, EGFR, HNSCC, uptake, PTT, NIR-II window

## Abstract

Gold nanoparticles have been indicated for use in a diagnostic and/or therapeutic role in several cancer types. The use of gold nanorods (AuNRs) with a surface plasmon resonance (SPR) in the second near-infrared II (NIR-II) optical window promises deeper anatomical penetration through increased maximum permissible exposure and lower optical attenuation. In this study, the targeting and therapeutic efficiency of anti-epidermal growth factor receptor (EGFR)-antibody-functionalised AuNRs with an SPR at 1064 nm was evaluated in vitro. Four cell lines, KYSE-30, CAL-27, Hep-G2 and MCF-7, which either over- or under-expressed EGFR, were used once confirmed by flow cytometry and immunofluorescence. Optical microscopy demonstrated a significant difference (*p* < 0.0001) between targeted AuNRs (tAuNRs) and untargeted AuNRs (uAuNRs) in all four cancer cell lines. This study demonstrated that anti-EGFR functionalisation significantly increased the association of tAuNRs with each EGFR-positive cancer cell. Considering this, the MTT assay showed that photothermal therapy (PTT) significantly increased cancer cell death (>97%) in head and neck cancer cell line CAL-27 using tAuNRs but not uAuNRs, apoptosis being the major mechanism of cell death. This successful targeting and therapeutic outcome highlight the future use of tAuNRs for molecular photoacoustic imaging or tumour treatment through plasmonic photothermal therapy.

## 1. Introduction

Head and neck squamous cell carcinoma (HNSCC) is one of the most common cancers worldwide and contributes to 90% of all head and neck cancers [1]. Alcohol, tobacco and human papilloma virus (HPV) infection are major known risk factors of HNSCC [2]. Although the current standard of therapy for HNSCC patients has an overall 5-year survival rate of 35–54% [3], these conventional therapies suffer from debilitating side effects, such as temporary or permanent loss of speech, loss of hearing, chewing and/or swallowing, fatigue, hair loss, sore throat following damage to healthy tissues of the throat, salivary gland, thyroid gland and lymph nodes [4,5,6,7]. The overall patient survival also depends on the stage and molecular subtypes of the HNSCC. Chung et al. [8] and Walter et al. [9] identified four different molecular subtypes of HNSCC with different degrees of recurrence-free-survival using gene expression techniques and showed that the subtype with high expression of epidermal growth factor receptor (EGFR) had the worse outcome. As most cancers are diagnosed at an advanced stage, the outcome of conventional therapies, such as surgery, radiotherapy and chemotherapy, is mostly disappointing. This is because resection cannot remove all tumours or because of a very narrow therapeutic window using a high dose of chemo- and radiotherapy to try to eliminate the tumour [10]. Targeted therapeutic approaches using the cetuximab monoclonal antibody to block the EGFR pathway largely failed to improve overall survival in many pre-clinical and clinical trials [11].

Suppression of apoptosis is one of the molecular hallmarks of cancer [12], and hence, inducing the apoptosis pathway will increase cancer cell death. Resistance to therapy is mainly due to overexpression of anti-apoptotic proteins of the Bcl-2 family (Bcl-2, BclxL, Bcl-w, Mcl-1) or the inhibitors of apoptosis proteins (survivin or XIAP) and downregulation of pro-apoptotic proteins (Bax, caspase 8, death receptors, p53/p73/p21 wafI) [13]. Due to their non-invasive nature, fewer adverse effects and high efficacy, the use of phototherapies, such as photothermal therapy (PTT), photodynamic therapy (PDT) and photoimmunotherapy (PIT), is being intensively studied. In particular, PTT and PDT have shown an excellent tumour ablation potential by generating heat and reactive oxygen species, respectively, that cause apoptosis of the tumour cells [14,15]. However, unlike PDT, PTT is not affected by a lack of oxygen in the target tissue, which is dominated by hypoxia, and hence, under such cases, PTT is favoured over PDT [15,16]. Therefore, PTT is a relatively new therapeutic strategy whereby contrast nanomaterials absorb light in the near-infrared (NIR) window and convert it to heat in order to selectively ablate cancerous tissues thermally [17,18,19].

Today, a variety of photothermal agents, such as noble metals, transition metal chalcogenides and oxides, carbon-based nanomaterials (e.g., carbon nanotubes, graphene), organic materials, carbon-based materials, protein-based materials, lipids and others, are widely used in cancer theranostics research [15,20]. It is suggested that particles with an SPR in the NIR-II optical window have a deeper tissue penetration and better sensitivity for early detection of tumour cells than those within NIR-I wavelength and visible light regions [21,22,23]. Therefore, this deeper penetration ability alongside minimal interaction with biological tissues, such as skin, fat, oxy- and deoxyhaemoglobin, an enhanced signal to noise ratio and a large maximum permissible exposure, make NIR-II a promising alternative to NIR-I for cancer imaging and ablation [24,25]. Gold nanoparticles (AuNPs), manufactured with a wide range of absorption spectrum in the NIR window [26,27], not only have an increased laser penetration due to increased tissue transparency to the electromagnetic spectrum (termed ‘the optical window’) but also increase the affordability of this theranostic agent due to its usage of a relatively low laser power [28]. In addition, due to their excellent physical and chemical properties of stability and biocompatibility, strong optical absorbance through SPR properties and suitability for surface functionalisation, gold nanoparticles of various size and shape have potential for widespread use in the detection and treatment of cancers [29,30].

Gold nanoparticle shapes include spherical, nanorods, nanoshells, nanocages, tripods and tetrapods [31]. Jain and colleagues showed that although nanoshells and nanorods have a NIR plasmon resonance wavelength, which increases with increased diameter, size-normalised nanorods had a linear relationship with their aspect ratio (AR) and produced a great efficiency in diagnosis and therapy compared with both nanospheres and nanoshells [32]. Furthermore, small-sized AuNRs, compared with large-size AuNRs (with similar AR), have a much higher thermal stability and increased photoacoustic imaging signal in the NIR-II region [27]. Hence, AuNRs with an excitation wavelength at 1064 nm SPR in the NIR-II region have a stronger penetration of biological media and imaging quality compared with those in the NIR-I window [33]. In order to increase their targeting efficiency and circulation half-life, the surface of nanoparticles is mostly modified by coating with functionalised polymers, PEG chains and others [34,35,36].

Most functionalisation of antibodies on the surface of nanoparticles takes place through either adsorption (interaction of opposite charges including hydrophobic, electrostatic, hydrogen bonding and van der Waals attractive forces), covalent bonding (carbodiimide chemistry (antibodies’ amine group, –NHS), maleimide chemistry (antibodies sulfhydryl group, –SH) and “click chemistry” (through azide–alkyne, –EDC/NHS or maleimide–thiol conjugation group) and adaptor molecules (via biotin– NeutrAvidin strong binding affinity) [37]. For instance, the mechanism of binding between AuNR and its common stabiliser, cetyltrimethylammonium bromide (CTAB), is electrostatic interaction, where biomolecules can directly bind to CTAB through either direct electrostatic adsorption or by physiosorption or covalent attachment of charged polymers, or replacement techniques where CTAB is first replaced by bifunctional linker (binding through –SH and –COOH terminal group) or thiolated molecules and then conjugated to moieties [38,39].

Although nanoparticles are ideal for theranostics, surface coating/functionalisation is the single most common cause of cytotoxicity. In addition, positively charged nanoparticles are also toxic compared with negatively charged nanoparticles, and those with a neutral surface charge are mostly biocompatible [31]. Several in vitro experiments showed that CTAB-coated AuNRs are toxic to cancer cells compared with polystyrene sulfonate (PSS) or polyallylamine hydrochloride (PAH)-coated AuNRs [40]. Furthermore, CTAB-coated AuNRs were found to be more toxic to human erythrocytes than polyethylene glycol (PEG)-coated AuNRs [41,42]. NeutrAvidin-, avidin- or streptavidin-coated nanoparticles are commonly used in nanotechnology as they have a high affinity for biotinylated probes including antibodies. In addition to its high affinity for biotin, NeutrAvidin has a low non-specific protein–protein interaction due to its neutral isoelectric point and has a high cellular uptake [43]. Overall, many cells survive short-term exposure to a low dose (<10 µg/mL) of nanoparticles with toxicity increasing in a dose- and time-dependent manner following particle internalisation and production of reactive oxygen species [31].

Despite their potential for increased tissue penetration for imaging and/or therapy, there is still little understanding of the long term toxicity, biocompatibility and tissue clearance of 1064 nm SPR AuNRs [44]. To date, cancer-targeted therapy using PTT remains an unmet challenge and there is a paucity of research conducted using AuNRs in the NIR-II window. We used commercially available NeutrAvidin-coated 10 nm × 67 nm AuNRs with 1064 nm SPR. Therefore, we compared the specific targeting potential of cetuximab (an anti-EGFR antibody) functionalised AuNR in both EGFR-positive and -negative cancer cell lines and evaluated in vitro the therapeutic potential of PTT in aggressive HNSCC cells.

## 2. Materials and Methods

### 2.1. Cell Culture

CAL-27 (ATCC CRL-2095) and MCF-7 (ATCC HTB-22) were cultured in DMEM (#31966-021, Gibco, Loughborough, UK with 10% (*v/v*) FBS (#F7524, Sigma, Gillingham, UK). KSYE-30 was grown in a 1:1 ratio of RPMI 1640 (#61870, Gibco) and Ham’s F12 (#31765-027, Gibco) supplemented with 2 mM glutamine (#25030-081, Gibco) and 2% (*v/v*) FBS. Hep G2 cells (ATCC HB-8065) were cultured in RPMI 1640 (#61870, Gibco) with 10% (*v/v*) FBS. All cells were passaged when 70–80% confluent and incubated at 37 °C in a humidified incubator under 5% CO_2_.

### 2.2. Flow Cytometry

Cell suspensions were incubated for 2 h at 37 °C to allow for receptor recycling following trypsinisation and then centrifuged at 400× *g* for 5 min. Then, 1 × 10^6^ cells in 100 µL FACS buffer (10% FBS in PBS) were blocked with 5 µL Human TruStain FcX (#422301, BioLegend, San Diego, CA, USA) for 10 min on ice. The cells were incubated with either 0.05 µg of recombinant monoclonal human EGFR (Research Grade Cetuximab Biosimilar, #FAB9577B, R&D Systems, Minneapolis, MN, USA) or of mouse IgG1 isotype (#IC002B, R&D Systems) biotinylated antibodies for 30 min on ice. Following washing, streptavidin–FITC secondary antibody (#F0030, R&D systems) was added at a concentration of 10 µL/10^6^ cells (10 µg/mL) and incubated with the cells for 30 min on ice in a dark place. Cells were washed and then analysed on an Attune Flow Cytometer (Applied Biosystems, Temecula, CA, USA) and the results were analysed using the Attune software (Applied Biosystems, Life Technologies, Waltham, MA, USA). The percentage of EGFR-positive cells, mean and median fluorescent intensities were determined for all the four cancer cell types stained using the anti-EGFR antibody compared with the isotype control.

### 2.3. Immunofluorescence Imaging (IF)

Cells were plated onto glass coverslips and allowed to grow to 70% confluence. Cells were washed with PBS and fixed in 4% PFA for 10 min at RT. The wells were further washed twice in PBS for 5 min each time and blocked in 10% (*v/v*) FBS for 1 h. Cells were then incubated with 1 µg/mL of primary anti-human EGFR goat polyclonal antibody (#AF231, R&D Systems) or a biotinylated anti-human IgG1 rabbit monoclonal antibody (#31-1019-02, 2BScientific) as a control in a humidified chamber for 1 h at RT. FITC rabbit anti-goat secondary antibody (#31509, Invitrogen) or streptavidin–FITC was added following 3 washes of 5 min each in PBS and counterstained using a Prolong Gold Antifade Mount nuclear stain (#P36930, ThermoFisher Scientific, Eugene, OR, USA). The stained cells were imaged using a Zeiss microscope (Jena, Germany) using filters for DAPI (acquisition time, 20 ms) and FITC (175 ms).

### 2.4. Gold Nanorods (AuNRs)

NeutrAvidin-coated 10 nm × 67 nm gold nanorods (#C12-10-1064-TN-PBS-50-1) with a 6.7 aspect ratio (AR) were purchased from Nanopartz (Loveland, CO, USA). The stock solution was at a concentration of 2.7 mg/mL (47 nM or 2.8 × 10^13^ nps/mL) with optical density (OD) of 78. The AuNRs had a localised longitudinal and transverse surface plasmon resonance (SPR) at 1064 and 506 nm, respectively. The concentrations of EGFR-targeted and untargeted AuNRs were determined using Genesys20 ultraviolet–visible–NIR spectroscopy (UV–vis–NIR) analysis (#4001-0, ThermoFisher Scientific) based on OD/mL according to the AuNRs manufacturer’s recommendation. For the untargeted group, 2 µL of the stock solution was diluted in 198 µL of PBS in a sterile Eppendorf tube (E1415-1510, Star Lab, Milton Keynes, UK), following 5 min sonication in an unheated water bath and 30 s of vigorous vortexing of the stock AuNRs. The diluted AuNRs were transferred into micro quartz cuvette tubes (CV10Q700, Thorlabs, Newton, MA, USA), and the absorbance values of AuNRs were measured following baseline determination using a blank sample (PBS) using Jenway visible spectrophotometry software according to the manufacturer’s protocol.

Similarly, the UV–vis absorbance values of targeted AuNRs were also determined after 2 µL of AuNRs was let to conjugate with 5.6 µL of biotinylated anti-EGFR (10 µg/mL) antibody in PBS by vortexing at 1000 RPM for 2 h at RT. This conjugation process of NeutrAvidin-coated AuNRs and biotinylated antibody resulted in the formation of a tAuNR conjugate (Figure 1A). NeutrAvidin-coated AuNRs were commercially sourced (Nanopartz) and their preparation chemistry (Au/polymer bridge/NeutrAvidin) remained proprietary; however, the binding of biotinylated anti-EGFR antibody to NeutrAvidin-coated AuNR was based on a strong biotin–NeutrAvidin binding affinity. The concentration of the AuNRs samples was calculated from the peak UV–vis–NIR absorbance values at 1064 nm after correcting for the dilution factor and subtracting the absorbance values of the blank sample [45]. Quantitation of targeted and untargeted AuNRs was repeated three times. All absorbance values at different wavelengths were divided by the maximum absorbance value at 400 nm wavelength in order to get extinction values [46], and the UV–vis analysis graph was plotted using normalised extinction (arbitrary units) and wavelength (nm). Compared with the uAuNRs, the absorbance values of tAuNRs UV–vis were significantly lower and this could be due to the antibody–AuNRs conjugation process involving 3 consecutive washes in PBS. Overall, a loss of 64.1%, 45.7% and 23.2% was calculated during 3 different time periods and an average total loss of 44.3% AuNRs was calculated following antibody conjugation process, and this loss was added as a compensation to all targeted groups. Figure 1B shows the normalised extinction values of both uAuNR and tAuNR at different wavelengths following three repeat measurements using UV–vis and their respective transmission electron microscopy (TEM) images (inset).

### 2.5. Cytotoxicity Assay

The cytotoxic effect of AuNRs on cancer cells was determined using a Cell Counting Kit-8 (CCK-8) assay (96992, Sigma-Aldrich). The cells were seeded in a 96-well plate (#3599, Corning) at 5 × 10^3^ cells/well for CAL-27 and Hep G2, and at 4 × 10^3^ for KYSE-30 and MCF-7 cells and incubated in a humidified incubator at 37 °C under 5% CO_2_ for 24 h. Treatment groups carried out per plate were medium only, cells only and uAuNRs only. After 24 h, serially diluted uAuNRs were added in 100 µL medium starting at a maximum concentration of 19 µg/mL (1.97 × 10^11^ nps/mL, 0.33 nM). After 72 h of incubation, 10 µL of CCK-8 was added to each well and incubated for another 3 h at 37 °C. Absorbance values of each well were read using a plate reader (#LB 940, Mithras, Berthold Technologies, Bad Wildbad, Germany) at 450 nm wavelength.

### 2.6. Quantification of AuNR Targeting

All four cancer cell lines were cultured in triplicate, on 22 mm × 22 mm glass coverslips, as described previously. Biotinylated anti-EGFR monoclonal antibody was conjugated with AuNRs as described above [24,25,26,47,48]. After culturing the cells for 24 h on coverslips, the supernatants were removed and either tAuNRs or uAuNRs in 2 mL of fresh complete medium were added into each well. A total of 1 × 10^11^ (9.67 µg, 0.17 nM) uAuNRs or tAuNRs were added per well, considering the average loss calculated (Figure 1) during the functionalisation process of the tAuNRs. Control wells received 2 mL of complete medium only. After 18 h of incubation, the membranes of the cells were visualised using MemBrite dye kit according to the manufacturer’s protocol (#30096T, Biotium, Fremont, USA) in a humidified incubator at 37 °C under 5% CO_2_ for 5 min. After washing, the cells were fixed in 4% (*v/v*) PFA for 10 min at RT. After 5 washes with PBS, the coverslips were mounted onto glass slides using Prolong Gold antifade mountant containing DAPI (#P36930, ThermoFischer Scientific). These slides were left overnight at RT in a dry and dark place. Dark-field images were taken using an inverted microscope (Nikon Eclipse Tie, Nikon UK Limited) with a 100× oil-coupled objective. Dark-field, DAPI (nuclear stain) and Texas Red (membrane stain) channels were used to acquire 5 images per coverslip, per cancer cell line for each condition. Illumination and camera exposure levels were kept consistent between each modality for all cell lines imaged. ImageJ (32-bit, 1997, Bethesda, MD, USA) was used to qualitatively quantify the level of optical scatter from AuNR populations which appeared as bright yellowish to yellow-to-orange colour AuNRs aggregates. The image contrast tool was used in ImageJ to reduce the background signal from cell, enhancing the optical scatter from AuNRs. All groups had the same level of signal processing to enable comparisons between cell lines and targeted/untargeted AuNRs. The nuclear marker DAPI and membrane stains were used as a guide to quantify the number of such AuNRs aggregates with clear signal intensity per cell. The numbers of tAuNRs and uAuNRs aggregates per cell with clear and strong optical scattering intensity were counted and compared using a two-tailed non-parametric Mann–Whitney test for each cell line of all the four cancer cells following a one-way ANOVA analysis. The total number of samples used for each analysis was indicated and the whole analysis was performed using GraphPad Prism software (8.1.0 (325), 2019, San Diego, CA, USA).

### 2.7. Photothermal Therapy

Photothermal therapy (PTT) experiments were carried out using both uAuNRs and tAuNRs in EGFR-positive cell line CAL-27 and EGFR-negative cell line MCF-7. The AuNRs used for these PTT experiments were purchased pre-functionalised, using either human anti-EGFR antibody, EGFR-hIgG1 (#DA12-10-1064-Ab-PBS-50-1, Nanopartz) or negative control polymer (#DA12-10-1064-NC-PBS-50-1, Nanopartz). Although these new AuNRs were different in the way they were functionalised from the AuNRs used in other experiments in this paper, both AUNRs were similar in size (10 nm × 67 nm), SPR (1064 nm), solvent (PBS), OD (50) and total volume (1 mL). The cytotoxic effect of these newly purchased pre-functionalised (tAuNRs) and negative control (uAuNRs) was determined using 3-(4,5-dimethylthiazol-2-yl)-2,5-diphenyl-2H-tetrazolium bromide (MTT) assay (#10133722, Invitrogen) 24 h following the 4 h incubation of cells with 0, 5, 10, 15, 20 and 25 µg/mL concentrations in 96-well plates. CAL-27 (8 × 10^3^/well) and MCF-7 (5 × 10^3^/well) cells were seeded in 96-well plates and incubated in a humidified incubator at 37 °C under 5% CO_2_. Each 96-well plate was set up in such a way that 4 different treatment groups were included per well: control, AuNRs only, AuNRs + laser and laser only. Media were removed when cells reached about 60–70% confluence and 20 µg/mL uAuNRs or tAuNRs was added to the AuNRs only and AuNRs + laser groups of CAL-27 and MCF-7 plates in a total volume of 100 µL and incubated at 37 °C under 5% CO_2_ for 4 h. Media containing the AuNRs were removed after 4 h as further incubation with AuNRs showed increased cell toxicity. Wells were washed using PBS to remove any unbound gold nanorods and fresh medium was added. Selected wells of the AuNRs + laser and laser only groups were irradiated using a continuous wave diode laser (B4-852-1500-15C, Sheaumann) at 1064 nm with a power density of 2 W/cm^2^ (10 mm beam diameter) for a total of 2 min each. The laser was mounted on a 3-axis motorised translation stage, as previously described [49], and alternate wells were irradiated to minimise bias of overheating of the next well from neighbouring well. Well temperature was recorded using infrared thermal imaging camera (#TIM 640, Micro-Epsilon Messtechnik GmbH & Co. KG). After exposure, the 96-well plates were incubated for 24 h, the medium was removed and a fresh medium containing 0.5 mg/mL MTT was added in 100 µL of medium and incubated for additional 3 h. After this, the MTT-containing medium was removed and 150 µL DMSO was added to assess cell viability based on the absorbance values of each well, using a plate reader at 620 nm wavelength.

### 2.8. Apoptosis and Necrosis

In order to detect and quantify cancer cell death following PTT, 8 × 10^3^ CAL-27 and 5 × 10^3^ MCF-7 cancer cells per well were seeded in 96-well plates using the same setup as that of PTT experiment above. Once cells reached confluence, the medium was removed, and tAuNRs (#DA12-10-1064-Ab-PBS-50-1, Nanopartz) were added at 20 µg/mL concentration in 100 µL fresh media into the ‘Au + laser’ and ‘Au only’ wells. The control and ‘laser only’ wells received fresh medium only. After 4 h of incubation at 37 °C under 5% CO_2_, the medium was removed, wells were washed once using 1× PBS and 100 µL fresh medium was added. The ‘Au + laser’ and ‘laser only’ groups were irradiated using 1064 nm laser with a power density of 2 W/cm^2^ for a total duration of 2 min each, as above. Following irradiation, the 96-well plates were incubated overnight at 37 °C under 5% CO_2_. The following day, the medium was removed, wells were washed once using PBS, trypsinised, and cells in the same group were pooled together and transferred to a 15 mL Falcon tube and centrifuged.

To quantify the number of apoptotic and necrotic cells, the FITC annexin V (#2330496, eBioscience, Carlsbad, CA, USA) and propidium iodide PI (#2290912, eBioscience) double staining kit (eBioscience) protocol was used. Briefly, cells were washed once in 1× PBS and then resuspended in 1× binding buffer (# 2290871, eBioscience) at 5 × 10^6^ cells/mL and 100 µL of the suspension was transferred to FACS tubes. A volume of 5 µL of FITC-conjugated annexin V was added to each tube and incubated for 15 min at RT. Cells were washed in 1× binding buffer, resuspended in 200 µL buffer and incubated with 5 µL PI. Unstained cells, annexin-V-only- and PI-only-stained cells were also used as a control. Samples were analysed using a CytoFLEX S (Beckman Coulter, Indianapolis, IN, USA), and flow cytometry data were quantified using the CytExpert V.2.4 software (2.4.0.28, 2021, Indianapolis, IN, USA).

## 3. Results

### 3.1. EGFR Is Highly Expressed in HNSCC Cell Lines

Flow cytometry analysis showed that CAL-27 and KYSE-30 had an increased fluorescence intensity signal, as shown by the right shift of the histograms, when stained for anti-EGFR antibody compared with that of isotype. No such shifts were observed for the EGFR-negative cell lines Hep G2 and MCF-7 (Figure 2A). Overall, while 67.7% and 80.54% of the total cell counts were positive for anti-EGFR staining in KYSE-30 and CAL-27, respectively, only 5.7% and 2.3% were positive in Hep G2 and MCF-7 cells, respectively. The median fluorescent intensity (MFI) of EGFR also showed an 8.7- and 7-fold increase for KYSE-30 and CAL-27 cells, respectively, compared with isotype. The EGFR-negative cell lines Hep G2 and MCF-7 had an MFI increase of only 1.1-fold (Figure 2B). The percentage total count and MFI results imply a significantly increased EGFR expression level in KYSE-30 and CAL-27 compared with Hep G2 and MCF-7 cells. This was confirmed by immunofluorescence staining. This demonstrated that the anti-EGFR antibody targeted AuNRs were internalised strongly by KYSE-30 and CAL-27 cancer cells compared with the Hep G2 and MCF-7 cancer cell lines (Figure 2C). While the KYSE-30 and CAL-27 showed a strong and specific extracellular membrane positivity for anti-EGFR antibody, there was no such reactivity measured in the Hep G2 and MCF-7 cancer cell lines.

### 3.2. AuNRs Cytotoxicity

The result of cell toxicity study using untargeted AuNRs showed very little cell death even at the maximum concentration of 19 µg/mL. At this concentration, 72 ±8.3%, 79 ±4.3%, 67 ±12.2% and 101 ±8.2% of KYSE-30, CAL-27, Hep G2 and MCF-7 cancer cells were viable, respectively (Figure 3). It was not possible to achieve a 50 % cell death with this maximum concentration dose of AuNRs, and hence an IC_50_ for these particles was not obtained. Therefore, the 95% confidence interval (CI) was used to describe the AuNRs concentration required to cause 50% cell toxicity. Hence, based on these data, the IC_50_ values were therefore extrapolated to be over 100 µg/mL for each cell line.

### 3.3. In Vitro Targeting Efficiency

To identify if there were any differences in cellular tAuNRs uptake in EGFR-positive KYSE-30 and CAL-27 cancer cells, the number of AuNRs aggregates associated with each cell following co-incubation was quantified based on small dots of clearly visible optical scatters from such AuNRs. Dark-field images indicated a higher AuNRs optical scattering in tAuNRs compared with uAuNRs (Figure 4A). A two-tailed nonparametric Mann–Whitney *t*-test showed a significant difference (*p* < 0.0001) between targeted and untargeted KYSE-30, indicating that antibody functionalisation increased the number of tAuNRs aggregates with optical scatter per cell (Figure 5A). The same was observed for the second EGFR-positive cancer cell line CAL-27 with a significantly higher (*p* < 0.0001) number of AuNRs aggregates per cell in the targeted than in the untargeted AuNRs (Figure 5B).

When EGFR-negative cancer cell lines Hep G2 and MCF-7 were imaged under dark-field microscope, a low number of AuNRs aggregates with bright signal scatters was observed (Figure 4B). The number of such AuNRs aggregates with optical scatter in Hep G2 cells indicated that there was a significantly higher (*p* = 0.006) AuNRs signal scatter per cell in the targeted group than in the untargeted group (Figure 5C). However, overall, this was insignificant compared with EGFR-positive cancer cell lines, with a median of zero and four AuNRs aggregates per cell for untargeted and targeted groups, respectively. When the second EGFR-negative cell line MCF-7 was analysed for the number of AuNRs aggregates per cell, there was still a significantly higher (*p* < 0.0001) number of such AuNRs aggregates per cell in the targeted group than in the untargeted group (Figure 5D). Again, the median number of AuNRs in MCF-7 remained low as in Hep G2, with zero and two aggregates of AuNRs optical scatters per cell for untargeted and targeted groups, respectively.

There was a significantly higher (*p* < 0.0001) number of tAuNRs aggregates in KYSE-30 compared with CAL-27, Hep G2 and MCF-7, suggesting that tAuNRs targeted KYSE-30 at an increasingly higher rate than they did the remaining three cancer cell lines (Figure 5E). Likewise, there was a significantly higher (*p* < 0.0001) number of tAuNRs aggregates per cell in CAL-27 than in both Hep G2 and MCF-7 cancer cells. When the ratios of the median numbers of AuNRs aggregates in tAuNRs and uAuNRs were compared for each cell line, they were found to be 6.4, 9, 4 and 2 for KYSE-30, CAL-27, Hep G2 and MCF-7 cells, respectively.

### 3.4. NIR-II Photothermal Therapy (PTT) Significantly Increased Cancer Cell Death in Head and Neck Cancer

The cell proliferation and viability assay after 4 h of incubation with AuNRs showed that increasing the concentration of both uAuNRs and tAuNRs gradually decreased cell viability in CAL-27 and MCF-7 cancer cell lines (Figure 6A). Overall, there was no major cytotoxicity difference between tAuNRs and uAuNRs in both cell lines. However, while uAuNRs tended to be more toxic to CAL-27 cells, particularly at 10 and 15 µg/mL, tAuNRs appeared to be slightly more toxic to MCF-7 at concentrations of 5 and 25 µg/mL. In this PTT experiment, a concentration of 20 µg /mL of uAuNRs or tAuNRs was used on both CAL-27 and MCF-7 cancer cell lines. PTT using a combination of uAuNRs and a 1064 nm laser at 2 W/cm^2^ for 2 min did not cause any significant decrease in cell viability above that observed for uAuNRs alone in the CAL-27 cancer cell line (Figure 6B). However, when tAuNRs were used in combination with the 1064 nm laser, the result showed a significant (*p* < 0.0001) increase in cell death compared with control, ‘laser only’ and ‘tAuNR only’ therapies. When the EGFR-negative MCF-7 cancer cells were treated in the same way, there was no significant difference in cell viability between uAuNRs alone and uAuNRs with the 1064 nm laser. However, the viability of MCF-7 cells in both ‘uAuNRs only’ and ‘uAuNRs and 1064 nm laser’ groups was significantly lower than that of control and ‘laser only’ treatment groups. More importantly, unlike the EGFR-positive CAL-27 cells, PTT did not result in as great a reduction in cell viability in EGFR-negative MCF-7 cells (59 ± 5% cells viable compared with 2.4 ± 3.2% viability in that of CAL-27 cells, Figure 6B) indicating the key role of targeting in PTT. Analysis of well-averaged temperature per different treatment groups in both CAL-27 and MCF-7 cultured plates showed a significant rise of temperature in CAL-27 only following the ‘tAuNRs + laser’ therapy (Figure 6C).

### 3.5. Combination of tAuNRs and Laser Therapy Caused Cell Death by Apoptosis

Necrosis and apoptosis are the two most common forms of cell death, whereby the former is uncoordinated cell death leading to release of content and activation of inflammatory response, while the latter is a controlled and preferred form of cell death ending up with removal of the content of the dying cells [50]. Only 2% of CAL-27 control cells (untreated) were annexin-V-positive (apoptotic) and this percentage did not rise when these cells were treated with tAuNR only or laser only (Figure 7). However, the combination of tAuNRs and laser irradiation increased early apoptosis of this cancer cell line to almost 11%. Although there was a slight increase in late apoptosis due to tAuNRs and laser only, the combination of the two resulted in 37% of cells in late apoptosis, with an additional 7% of cells in necrosis. There was a far smaller number of cells encountered during analysis in the ‘tAuNRs + laser’ group of CAL-27 cells following combination therapy, probably due to detachment. The EGFR-negative MCF-7 cells showed a slight increase in early apoptotic cells when treated with tAuNR (6.11% of cells) or laser only (6.34% of cells) compared with the control group (2.21% of cells). However, unlike in CAL-27 cells, the combination therapy did not increase early apoptosis (2.86% of cells). The proportion of MCF-7 cells in late apoptosis was similar (6.66–7.34%) for control, ‘laser only’ and ‘combination therapy’ treatment groups, whereas tAuNRs caused a slight increase to 12.58% of cells in late apoptosis (Figure 8). The overall mean number of cells in early and late apoptosis for MCF-7 cancer cell was 10.2%, which was quite close to that of the percentage of necrotic-stage cell in the control (9.57%), tAuNRs (9.8%) and ‘laser only’ (8.84%) groups. To summarise, while the combination of tAuNRs with laser caused increased cell death by apoptosis in the EGFR-positive cancer cell line, CAL-27, this was not observed for the EGFR-negative control cell line, MCF-7.

## 4. Discussion

This in vitro study identified that anti-EGFR functionalisation increased the targeting potential of the AuNRs to EGFR-expressing cell lines. These findings are supported by other studies for different cancer types, including colorectal cancer [51], cervical cancer [52] and skin cancer [53], where each of them found an increased uptake of functionalised gold nanoparticles in EGFR-expressing cancer cells in in vitro studies. In our study, although we used two cancer cell lines with an almost equal level of EGFR expression, as shown by flow cytometry (Figure 2), the number of tAuNRs detected in KYSE-30 was significantly greater (*p* = 0.002) than the one of tAuNRs identified in CAL-27 cancer cells. This seems to be due to an increased non-specific uptake of AuNRs. The exact reason for these differences between cell lines with both polygonal shape and similar EGFR expression is not clear. However, examination of these cells under the microscope, at 100× magnification, showed that KYSE-30 cells were two-fold the size of CAL-27 and had more filopodia on their cell membrane than the CAL-27 cancer cells. Thus, these differences in uptake could be attributed to the differences in intracellular signalling cascade of these cells. A study by Khaznadar et al. showed that despite both KYSE-30 and CAL-27 having comparable level of EGFR expression, the level of EGFR phosphorylation in KYSE-30 was more than double that of CAL-27 cancer cells [54]. Phuc et al. also found an increased efficiency of gold nanoparticles internalisation in EGFR-expressing A431 cells through clathrin-mediated endocytosis in the presence of EGF, and the absence of such ligands dramatically reduced the uptake of these nanoparticles [55].

The present findings of a significantly increased number of internalised tAuNRs aggregates with optical scatter in the EGFR-positive cells compared with uAuNRs indicate the importance of targeting. However, the presence of a relatively higher number of tAuNRs than uAuNRs in EGFR-negative cancer cells may be due to a change of surface charge on AuNRs following functionalisation, which might affect their internalisation. When the highly EGFR-expressing lung cancer cell line HCC87 and EGFR-negative human kidney 293T cells were compared for uptake of peptide-conjugated triangular gold nanoplates, there was a significantly higher internalisation in the HCC87 than in the 293T cells [56]. Similarly, Li and colleagues also compared the relative uptake of cetuximab-functionalised gold nanoparticles (AuNPs) by EGFR-expressing A431 and EGFR-negative MDA-MB-453 cancer cells, using transmission electron microscopy (TEM); they found more AuNPs bound and accumulated in A431 cancer cells [53].

Specific targeting of nanoparticles to improve therapy has long been a goal of many nanomedicine research studies. In our PTT study, the MTT assay showed that the use of tAuNRs at 20 µg/mL concentration significantly increased cancer cell death (97.6 ± 3.2%) in the EGFR-positive cancer cell line CAL-27, following a significant rise in a well-averaged temperature. This finding is in agreement with that of Yang et al., who showed increased cancer cell death (~77.5%) using 50 µg/mL concentration of lactoferrin-functionalised AuNRs to target Hep G2 cancer cells and irradiated them using a 980 nm diode laser at a power density of 0.5 W/cm^2^ for 1 min [57]. Another in vitro PTT study, which used 30 µg/mL concertation of anti-EGFR functionalised AuNRs in combination with a 808 nm laser at 2 W/cm^2^ power density for 5 min in MDA-MB-231 cancer cell, demonstrated around 76.5% of cancer cell death [58]. Although some differences exist between these two studies and ours, including power density and duration of therapy, it is important to notice that targeting AuNRs generally increased the outcome of PTT. Overall, targeting may play a useful role in improving the PTT response of cancer cells, and such results also depend on the concentration of AuNRs, laser wavelength (NIR-I or NIR-II), laser power density and the duration of irradiation. As the result of any MTT assay shows only cell viability based on a measurement of the mitochondrial metabolic activity, an apoptosis assay was employed to determine the mechanism behind this reduction in cell viability following PTT. Using annexin V and PI double staining, we demonstrated that a combination of tAuNRs and laser irradiation increased the percentage of cells in apoptosis to approximately 48% in EGFR-positive CAL-27 cells. When the same combination therapy was used in the EGFR-negative cancer cell line MCF-7, only around 10% of cancer cells underwent apoptosis. This result demonstrates that there is a huge potential for targeting and selectively ablating cancer cells using PTT, which induces apoptosis in such targeted cells.

One of the probable obstacles to the use of nanoparticles in cancer therapy is their cell and tissue toxicity [59,60]. A wide range of in vitro experiments showed that colloidal particles in the range of 3–100 nm in size have no evident toxicity as long as the 10^12^ nps/mL threshold is not exceeded, and an in vivo experiment also showed a lack of observable short term toxicity at 0.5 mg/kg [61]. This shows our current 1.97 × 10^11^ nps/mL usage is well below this threshold. Various previous in vitro targeting studies reported effects of AuNR concentration, duration of incubation and surface functionalisation on viability of different cancer cell types. For example, Zhang et al. reported a minimal cytotoxic effect of AuNRs following incubation of MDA-MB-231 cells with 1.84 µg/mL of AuNRs for 24, 48 and 72 h; the presence of anti-EGFR or tAuNR showed some cytotoxic effect at 48 and 72 h time points [62]. In our study, NeutrAvidin-coated AuNRs caused a slight reduction in cell viability after 72 h incubation in KYSE-30 (72%), CAL-27 (79%) and Hep G2 (67%), but no cytotoxic effect on MCF-7 cells (101%). Given that the same size of AuNRs was used throughout this study, the slightly increased cell toxicity from the second batch was probably due to their surface coating. Kao et al. demonstrated a lack of any of such differences in cell viability when A549 cancer cells were incubated with either AuNPs–PEG or cetuximab–AuNPs–PEG during the first 24 h at a concentration of 0.625 µg/mL [63]. Another study using MDA-MB-231 cells and different concentrations of anti-EGFR-functionalised AuNRs at 24 and 48 h time points also showed reduced cell toxicity effect of AuNRs, even at a higher concentration of 100 µg/mL [58]. This agrees with our current findings, where a maximum concentration of 19 µg/mL for 72 h did not produce any significant reduction in viability in all four cancer cell lines used in this paper. Cell viability was reduced with the tAuNRs used in the PTT experiments, presumably due to their surface functionalisation of pre-conjugate targeting antibody or control polymer. However, as the surface coating remains proprietary knowledge of the company, it is beyond the scope of this paper to comment further on why this may be so.

In conclusion, this study showed that the tAuNRs with SPR at 1064 nm had an increased uptake in EGFR-expressing cell lines compared with untargeted particles. However, the amount of uptake was dependant on the cell line, as it was significantly higher in the KYSE-30 compared with CAL-27 cancer cells. We hypothesise that the increased number of AuNRs aggregates in EGFR-positive cancer cells was likely due to ligand–receptor binding activity [64] as opposed to simple passive diffusion mechanism. The work of Van Lehn et al. showed increased internalisation of AuNPs into HeLa cells when incubated at 37 °C compared with 4 °C, depending on their size and functionalisation [65], indicating the importance of active transport for the cell internalisation of nanoparticles. To the contrary, several older studies described the enhanced permeability retention (EPR) effect [66,67] as a mechanism of nanoparticles cell entry mechanism. However, a recent in vivo study [68] disputed this understanding of the EPR effect, claiming that it is an active endocytosis (either inside vesicles or through transcellular channels), not a passive diffusion, that plays a major role in the nanoparticles cell entry mechanism. Overall, the result of this in vitro demonstration of targeting of particles with strong light absorbance in the second optical window showed a successful cancer cell ablation using PTT due to its photothermal conversion efficiency than first optical window. This opens future potential for these AuNRs to be used as molecular targeting of light sensitisers for photoacoustic imaging and/or photothermal therapy in preclinical and clinical applications.

## Figures and Tables

**Figure 1 pharmaceutics-13-01651-f001:**
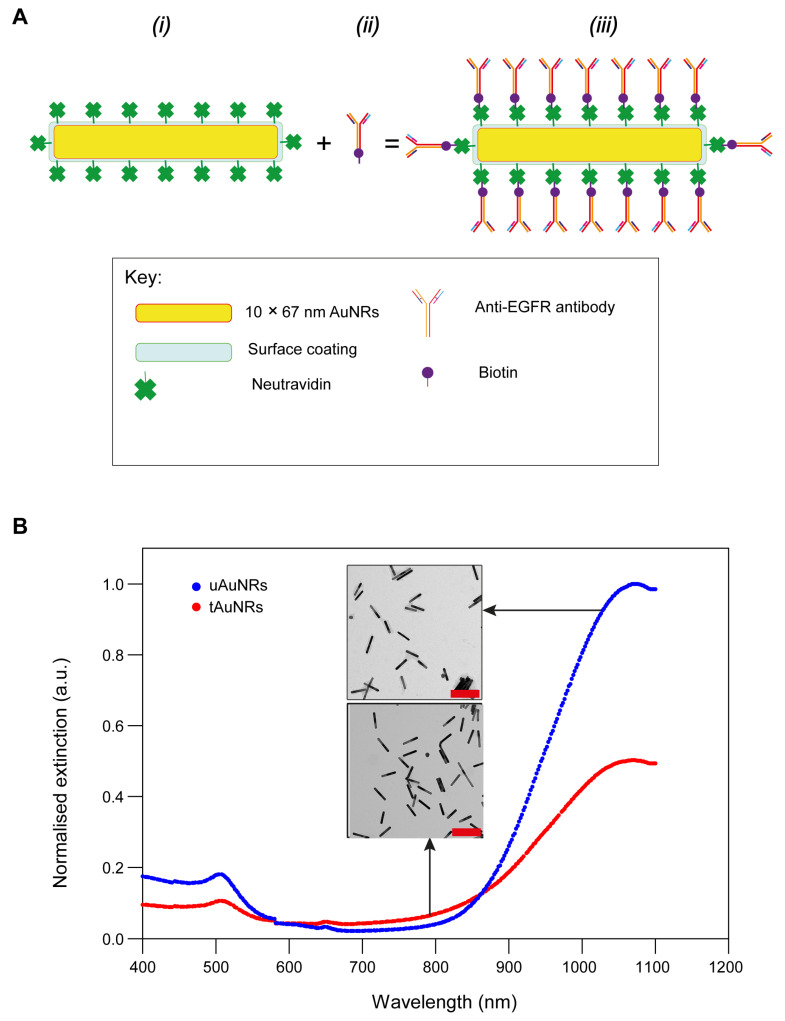
Antibody conjugation process reduced the number of AuNRs. (**A**) Schematic diagram showing a structure of tAuNRs. NeutrAvidin (4 nm × 5 nm × 5.6 nm)-coated10 nm × 67 nm AuNRs (***i***) were added to biotinylated (5.2 nm × 5.2 nm) Anti-EGFR antibody (***ii***) in a 1:2.8 ratio (*v/v*) and conjugated to form tAuNRs (***iii***). (**B**) Absorbance measurement was used to plot the normalised extinction values of targeted and untargeted AuNRs. A loss of 44.3% ± 20.5 of AuNRs in the conjugation process was calculated. TEM images (inset) of uAuNRs and tAuNRs; scale bar (red), 200 nm.

**Figure 2 pharmaceutics-13-01651-f002:**
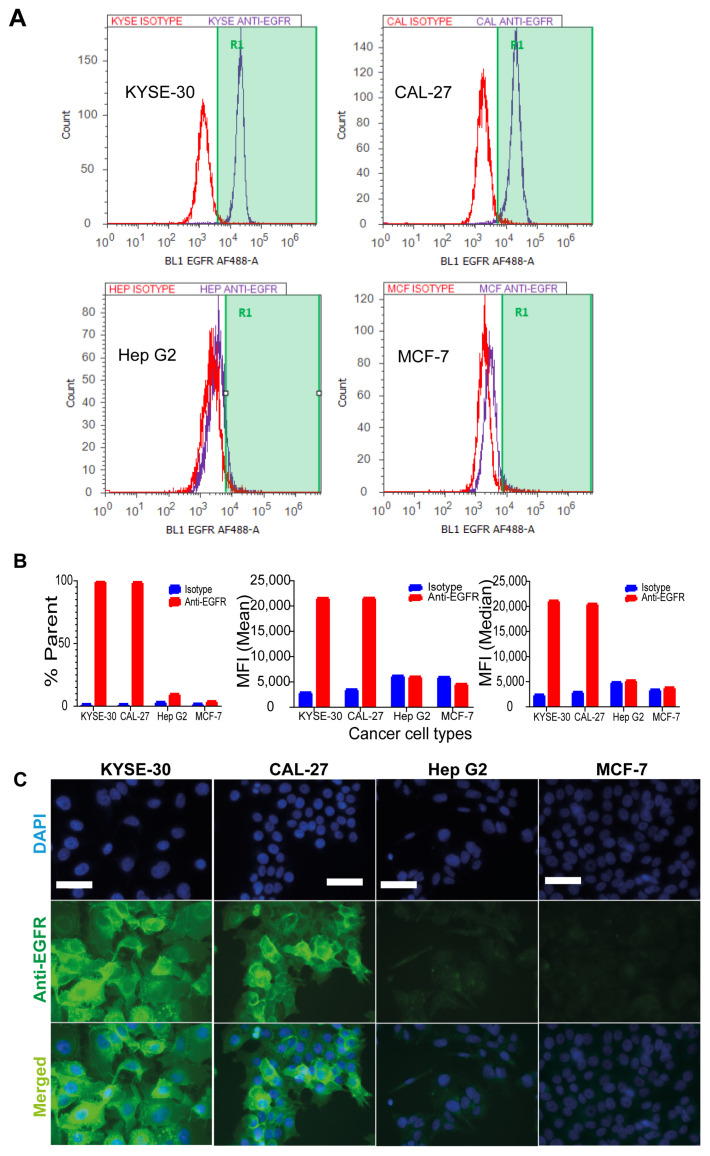
CAL-27 and KYSE-30 cancer cells express high level of EGFR. (**A**) Flow cytometry (red, isotype and blue, anti-EGFR staining); EGFR-positive cancer cell lines KYSE-30 and CAL-27 had an 8.7 and 7-fold increase in fluorescence in the presence of anti-EGFR antibody, respectively, while EGFR-negative cancer cell lines Hep G2 and MCF-7 showed no such increase. The *y*-axis represents the number of events (counts) and the *x*-axis represents the fluorescence signal intensity from the EGFR-bound antibody. (**B**) Quantification of flow cytometry results as a percentage of total cells (% parent), mean and median fluorescence intensity of both isotype and anti-EGFR antibodies in 4 different cancer cell types. (**C**) Immunofluorescence staining of these four cancer cell lines using anti-EGFR and isotype antibodies showed a high expression of EGFR in KYSE-30 and CAL-27 compared with Hep G2 and MCF-7 cancer cell lines. Scale bars, 50 µm.

**Figure 3 pharmaceutics-13-01651-f003:**
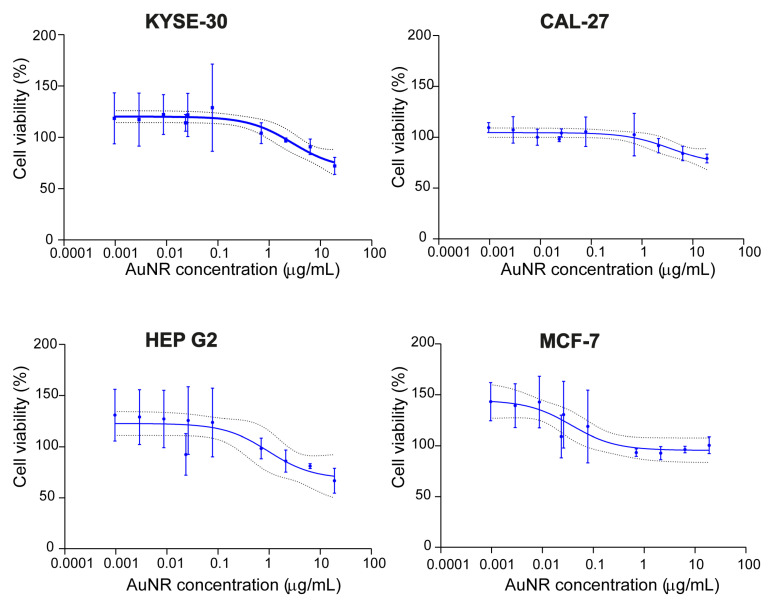
AuNRs had no cytotoxic effect on either EGFR-positive or EGFR-negative cancer cells. Dose-response curves for different cancer cell lines, *n* = 3, the dotted lines represent 95% CI.

**Figure 4 pharmaceutics-13-01651-f004:**
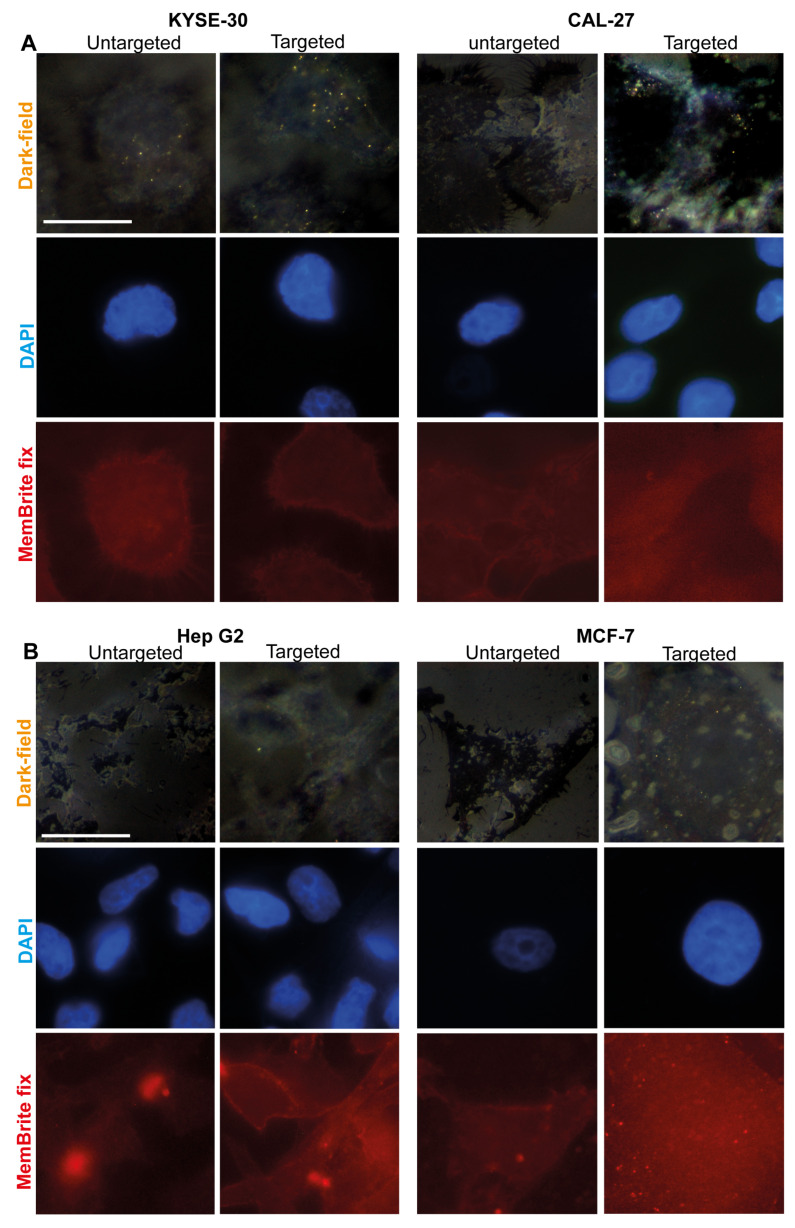
Images showing different intensities of AuNRs signals in EGFR-positive and negative cancer cell lines. Dark-field, DAPI and Texas Red images of (**A**) EGFR-positive cancer cell line KYSE-30 and CAL-27 and (**B**) EGFR-negative cancer cell lines Hep G2 and MCF-7 were taken for both tAuNRs and uAuNRs groups. All dark-field (where bright yellow-to-orange colour aggregates represent AuNRs), DAPI and Texas Red (MemBrite fix) images were captured using an inverted Nikon microscope with a uniform setting throughout. Scale bar, 200 µm.

**Figure 5 pharmaceutics-13-01651-f005:**
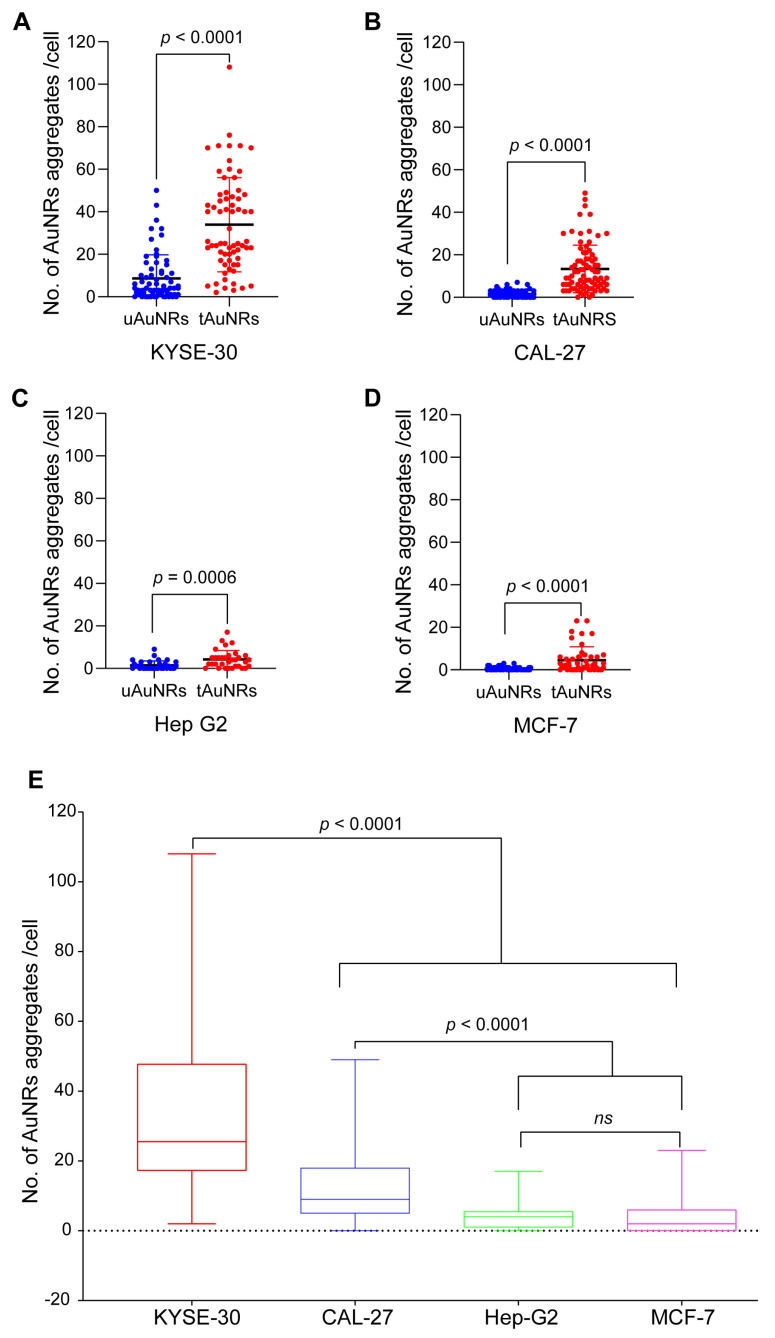
Targeting increased the number of AuNRs in cancer cells. (**A**–**D**) Dot plots show a qualitative quantification of AuNRs aggregates with a strong optical scatter from aggregates imaged and counted per cell. Mann–Whitney test showed that there was a significantly higher number of AuNRs aggregates counted in tAuNRs than in uAuNRs in: (**A**) KYSE-30 (*p* < 0.0001); (**B**) CAL-27 (*p* < 0.0001); (**C**) Hep G2 (*p* = 0.0006) and (**D**) MCF-7 (*p* < 0.0001). (**E**) Comparison of the number of tAuNRs aggregates per cell showed a significantly increased number of tAuNRs aggregates in EGFR-positive compared with EGFR-negative cancer cell lines. The total number (*n* = 459) of cells counted were: KYSE-30, 68 for each tAuNR and uAuNR; CAL-27, 68 for uAuNR and 83 for tAuNR; Hep G2, 33 for each uAuNR and tAuNR; MCF-7, 59 for uAuNR and 47 for tAuNR. Box plot shows median and range with their statistical significance where ns indicates ‘not significant’ statistical value.

**Figure 6 pharmaceutics-13-01651-f006:**
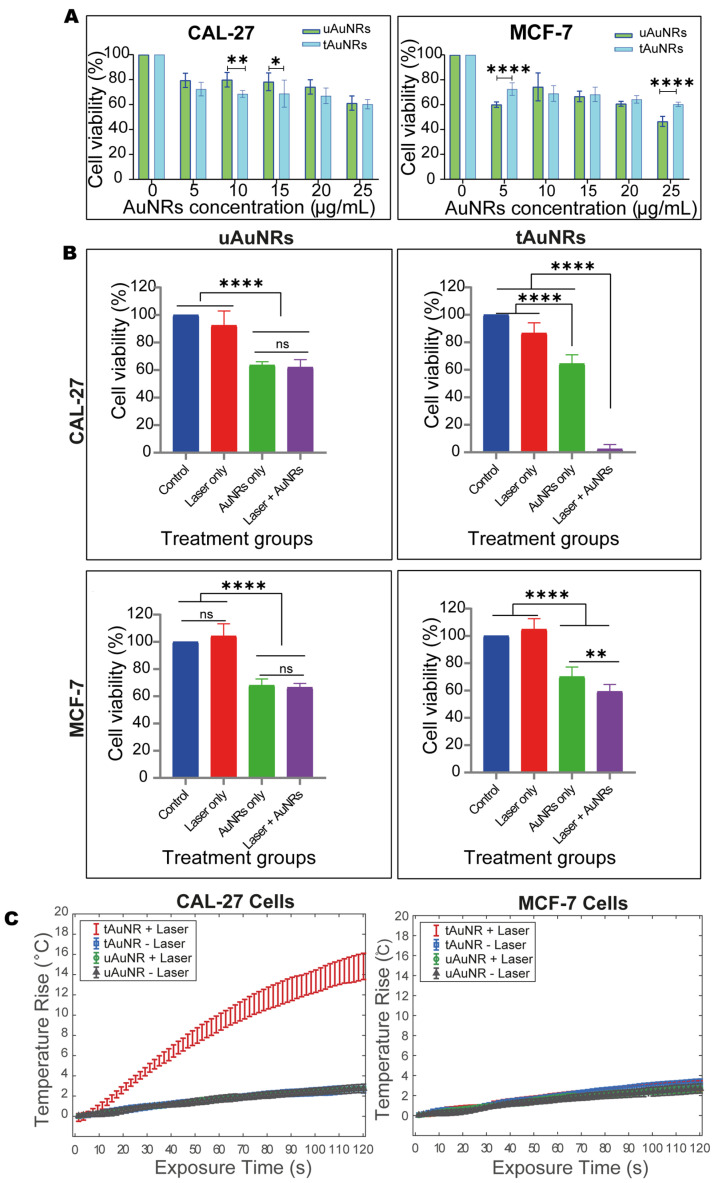
Photothermal therapy increases cancer cell death using EGFR-targeted AuNRs. (**A**) Analysis by ordinary two-way ANOVA showed reduced cell viability in both CAL-27 and MCF-7 cancer cells incubated with increasing concentrations of uAuNRs and tAuNRs. (**B**) Ordinary one-way ANOVA test demonstrated that there was no significant reduction in cell viability from the combination of uAuNRs with laser in both CAL-27 and MCF-7 cells other than the toxicity caused by the AuNRs themselves. On the other hand, tAuNRs and laser combination caused a significant reduction in cell viability in CAL-27 cancer cells and a relatively less significant reduction in cell viability in MCF-7 cells. (**C**) Well-averaged temperature rise from the different exposure conditions in CAL-27 and MCF-7 cancer cell lines. For all conditions, data are shown as mean ± SD and *n* = 7 or 8. Asterisks indicate a significant difference: * *p* < 0.5, ** *p* < 0.01, **** *p* < 0.0001 and ns, not significant.

**Figure 7 pharmaceutics-13-01651-f007:**
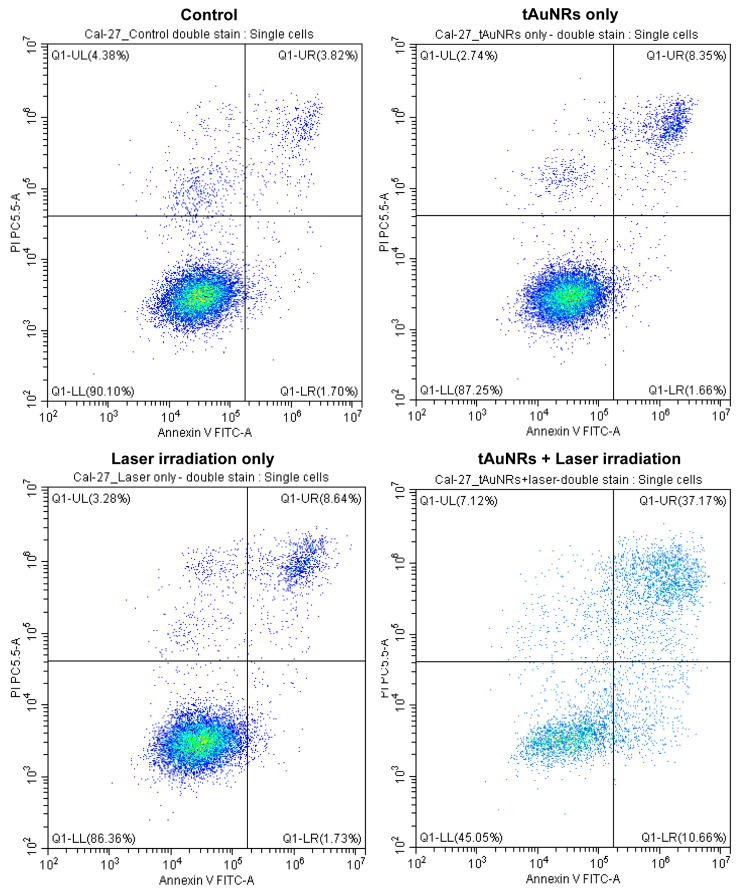
Apoptotic cell death in CAL-27 cells following tAuNR and laser treatment. Cells were incubated with tAuNRs alone, with laser irradiation alone or with their combination (tAuNRs + laser irradiation). Flow cytometry analysis of these cells stained with annexin V FITC and PI alongside untreated (control) cells is shown. Gating was determined based on the control cells and used for the treatment groups to determine the percentage of healthy cells (Q1-LL), cells in early apoptosis (annexin-V-positive, PI-negative, Q1-LR), cells in late apoptosis (annexin-V-positive, PI-positive, Q1-UR) and the percentage of cells undergoing necrosis (PI-positive, annexin-V-negative, Q1-UL). Total number of single cell events: control, 10,728; tAuNRs, 10,890; laser only, 10,988 and tAuNRs + laser, 4899.

**Figure 8 pharmaceutics-13-01651-f008:**
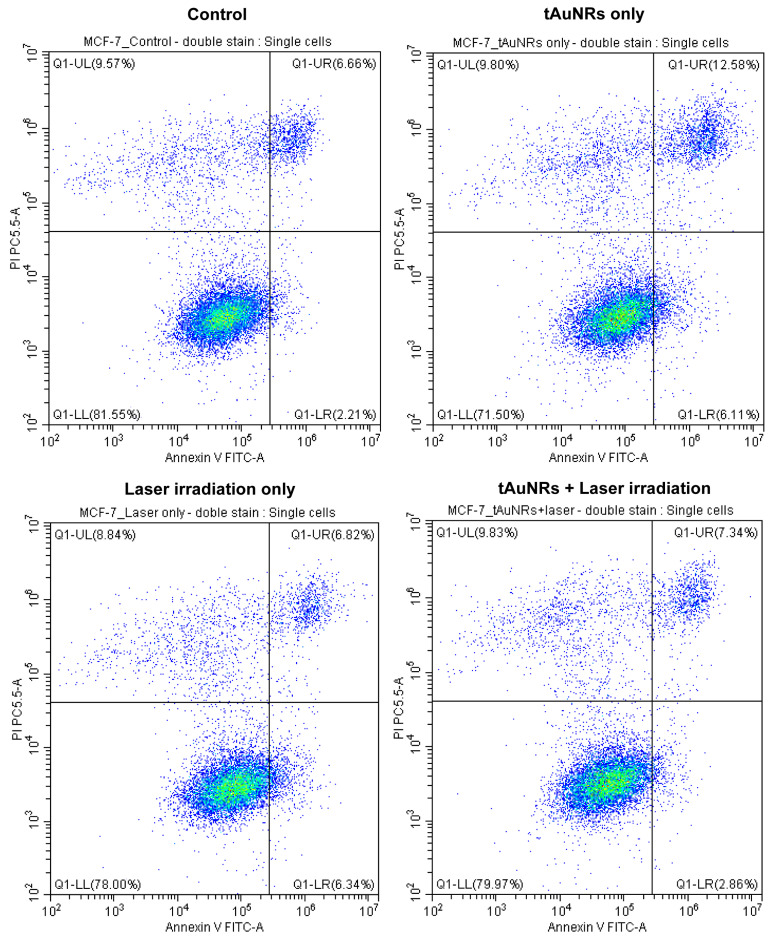
Apoptotic cell death in MCF-7 cells following tAuNR and laser treatment. Cells were incubated with tAuNRs alone, with laser irradiation alone or with their combination (tAuNRs + laser irradiation). Flow cytometry analysis of these cells stained with annexin V FITC and PI alongside untreated (control) cells is shown. Gating was determined based on the control cells and used for the treatment groups to determine the percentage of healthy cells (Q1-LL), cells in early apoptosis (annexin-V-positive, PI-negative, Q1-LR), cells in late apoptosis (annexin-V-positive, PI-positive, Q1-UR) and the percentage of cells undergoing necrosis (PI-positive, annexin-V-negative, Q1-UL). Total number of single cell events: control, 11,920; tAuNRs, 12,495; laser only, 11,775 and tAuNRs + laser, 12,079.

## Data Availability

Not applicable.

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
