# Peer review of "Evaluation of the Targeting and Therapeutic Efficiency of Anti-EGFR Functionalised Nanoparticles in Head and Neck Cancer Cells for Use in NIR-II Optical Window"

_pharmaceutics, 2021, doi:10.3390/pharmaceutics13101651_

Round 1

Reviewer 1 Report

Egnuni et al. investigated the targeting and therapeutic efficiency of anti-epidermal growth factor receptor (EGFR) antibody functionalised AuNRs with an SPR at 1064 nm in vitro. This is an interesting study. however, there a lot of unclear mechanisms which I have listed below;

  • Line 29 - is protein examples of NPs? please rewrite this sentences. Proteins and other biomolecules are of same size of NPs, but they are not NPs. Although there is a huge difference between engineered and biological NPs.
  • Authors did not explain why the optical windows of NIR-I and NIR-II are required. Of course they are highly desirable but they should be explained properly in the paper.
  • how AuNPs have become ideal candidate for cancer therapy. Authors should add that AuNPs are found in awide variety of shapes, and then they should explain why do they prefer nanorods. I suggest adding a reference here; https://doi.org/10.1002/advs.201903441
  • Authors should also explain why PTT is prefered over other therapeutic modailities, for example, PDT, chemotherapy, radiation therapy, surgery, immunotherapy, etc.
  • why did authors choose, Neutravidin coated 10 nm x 67 nm gold nanorods? and why these dimensions, all these aspects should be explained for readers. 
  • Scale bar in figure 1 is not readable. please add this.
  • The concentration of AuNRs should be explained, and how this range of concentration is relevant to clinical environment.
  • Scale bars in figure 2 and 4 are missing. please add them.

Author Response

Response to reviewers’ comments:

We thank the reviewers for their useful and insightful comments and suggestions on our work, and we hope to have addressed them with this revision. All changes have been highlighted yellow in the main text wherever possible and referred by their line number in response to the specific comment below.

Reviewer 1’s comments

  1. Line 29 - is protein examples of NPs? Please rewrite this sentences. Proteins and other biomolecules are of same size of NPs, but they are not NPs. Although there is a huge difference between engineered and biological NPs.

Response – Yes, proteins are not nanoparticles themselves and this sentence has been clarified. Please see line 63-66.

  1. Author didn’t explain why the optical windows of NIR-I and NIR-II are required. Of course they are highly desirable but they should be explained properly in the paper.

Response – The NIR-I and II region, dubbed ‘optical window’, has been chosen because it is important in PTT for its maximum permissible exposure, deep tissue/media penetration, minimum interaction with biological tissue and enhanced signal to noise ratio. Please see line 68-75 of the manuscript.

  1. How AuNPs have become ideal candidate for cancer therapy? Authors should add the AuNPs are found in a wide variety of shapes, and then they should explain why you prefer nanorods. I suggest adding a reference here; https://doi.org/10.1002/advs.201903441

Response – AuNPs are available in several shapes, and they can be tuned for therapy from visible light to NIR region. They do also have excellent physical and chemical properties of stability and biocompatibility, ease of surface functionalisation with targeting agents and have dual photoacoustic and photothermal therapy advantages. Particularly, AuNRs have a unique transverse and longitudinal surface plasmon resonance (LSPR) which can be tuned in the NIR region by changing their aspect ratio. Please see lines 80 – 90. We have added the suggested reference.  

  1. Authors should also explain why PTT is preferred over other therapeutic modalities, for example, PDT, chemotherapy, radiotherapy, surgery, immunotherapy, etc.

Response – PTT is one of the emerging cancer therapeutic modalities and it has been shown to treat cancer by generating heat in a localised region of body part and induce apoptosis without damaging the surrounding healthy tissue or cells. See lines 53-61.

  1. Why did authors choose neutravidin coated 10 nm x 67 nm gold nanorods? And why these dimensions, all these aspects should be explained for readers.

Response – AuNRs with a smaller diameter size have a stronger thermal stability compared to larger diameter sized nanorods. Hence, we selected a 10 nm x 67 nm AuNRs with 1064 nm SPR for both current in vitro as well as future in vivo experiment. Biotin-neutravidn interaction is one of the strongest non-covalent binding affinities and we employed this advantage to functionalise neutravidin coated AuNRs with our choice of biotinylated anti-EGFR antibody. Please see lines 84-90; 122-123.

  1. Scale bars in figure 1 is not readable, please add this.

Response – Scale bar font size and length has been increased to improve readability. Please see Figure 1 (lines 194 and 200).

  1. The concentration of AuNRs should be explained, and how this range of concentration is relevant to clinical environment.

Response – The concentration of the stock solution has been added to this manuscript in different units. Please see 162-163; 208. The current AuNRs concentration used in this study is well below the maximum threshold recommended to be used in in vitro study for clinical relevance. Please line 515-519.

  1. Scale bars in figure 2 and 4 are missing. Please add them.

Response – As the scale bars were small to see, I have changed both their size and weight. Please see Figure 2 (lines 307 and 308) and Figure 4 (lines 355 and 356).

Reviewer 2 Report

The manuscript entitled “Evaluation of the targeting and therapeutic efficiency of anti-EGFR functionalised nanoparticles in Head and Neck cancer cells for use in NIR-II optical window” discussed the targeting the use of tAuNRs for molecular photoacoustic imaging or tumour treatment through plasmonic photothermal therapy. This manuscript can be improved after major revision. The issues are listed as below.

  1. Author must include section on chemistry synthesis process other than just scheme. In this present version, proper preparation and functionalization process is missing.
  2. Author must more describe on biocompatibility and toxicity issue among this kind functionalized and their base nanoparticles on normal non-cancerous cell lines with respect to each cancer cell line used in the study.
  3. Author must include apoptosis specific death and expression cell death or apoptosis related markers and related pathways using PCR and western blotting after 24hr incubation. List of molecular markers involved must be included in the revised manuscript. Author can make separate figure for it.
  4. Figures 2, and 4 must be improved including font size inside figures.

Author Response

Response to reviewers’ comments:

We thank the reviewers for their useful and insightful comments and suggestions on our work, and we hope to have addressed them with this revision. All changes have been highlighted yellow in the main text wherever possible and referred by their line number in response to the specific comment below.

Reviewer 2’s comments

  1. Author must include section on chemistry synthesis process other than just scheme. In this present version, proper preparation and functionalisation is missing.

Response – Neutravidin coated AuNRs were commercially sourced (Nanopartz) and their preparation remained proprietary. However, as a background, a general nanoparticle synthesis methods have been included in this manuscript. Please see lines 93-103 and 177-180.

  1. Author must more describe on biocompatibility and toxicity issue among this kind functionalized and their base nanoparticles on normal non-cancerous cell lines with respect to each cancer cell line used in the study. [i.e are the base or functional group toxic or biocompatible?]

Response – One of the draw backs of the use of nanoparticles in nanomedicine is their poor biocompatibility or cytotoxicity. This study demonstrated low cell toxicity from the use of neutravidin coated AuNRs using MTT assay of 4 cancer cell lines. Following reviewers’ comment background information on biocompatibility and toxicity has been added to this manuscript. Please see lines 105-117; 513-516; 520-522; 532-534.

  1. Author must include apoptosis specific death and expression cell death or apoptosis related markers and related pathways using PCR and western blotting after 24 hr incubation. List of molecular markers involved must be included in the revised manuscript. Author can make separate figure for it.

Response – In our PTT experiment we clearly showed that significantly reduced cell viability was observed when we used a combination of tAuNRs and laser therapy in CAL-27 cancer cell line. However, there was no such increase in temperature using the uAuNRs for both CAL-27 and MCF-7. Hence, here we used tAuNRs on EGFR positive cancer cell line CAL-27 and the control EGFR negative MCF-7 cells. The mechanism of cell death following PTT was determined using flow cytometry analysis of Annexin V and PI-stained cells. Please see the sections of abstract (line 25), introduction (lines 49-53), materials and methods (268-289), result (416-436; Figures 7 & 8; 441-449; 451-458) and discussion (504-512).

  1. Figure 2, and 4 must be improved including font size inside figures.

Response – The font size and content of both Figure 2 and 4 has been improved, (Figure 2 – lines 307 and 318, Figure 4 – lines 355 and 362).

Round 2

Reviewer 1 Report

I am pleased to recommend the revised manuscript for publication in Pharmaceutics.

Reviewer 2 Report

I reccomend accepting this manuscript as author revised it properly